# Coupling of nanocrystal hexagonal array and two-dimensional metastable substrate boosts H$_2$-production

Zhenglong Fan[1,2,6], Fan Liao[1,6], Yujin Ji[1,6], Yang Liu ●[1] ✉, Hui Huang ●[1], Dan Wang ●[3], Kui Yin[1], Haiwei Yang[1], Mengjie Ma[1], Wenxiang Zhu[1], Meng Wang[1], Zhenhui Kang ●[1,4] ✉, Youyong Li ●[1], Mingwang Shao ●[1] ✉, Zhiwei Hu ●[5] ✉ & Qi Shao ●[2] ✉

Designing well-ordered nanocrystal arrays with subnanometre distances can provide promising materials for future nanoscale applications. However, the fabrication of aligned arrays with controllable accuracy in the subnanometre range with conventional lithography, template or self-assembly strategies faces many challenges. Here, we report a two-dimensional layered metastable oxide, trigonal phase rhodium oxide (space group, P-3m1 (164)), which provides a platform from which to construct well-ordered face-centred cubic rhodium nanocrystal arrays in a hexagonal pattern with an intersurface distance of only 0.5 nm. The coupling of the well-ordered rhodium array and metastable substrate in this catalyst triggers and improves hydrogen spillover, enhancing the acidic hydrogen evolution for H$_2$ production, which is essential for various clean energy-related devices. The catalyst achieves a low overpotential of only 9.8 mV at a current density of −10 mA cm$^{-2}$, a low Tafel slope of 24.0 mV dec$^{-1}$, and high stability under a high potential (vs. RHE) of −0.4 V (current density of ~750 mA cm$^{-2}$). This work highlights the important role of metastable materials in the design of advanced materials to achieve high-performance catalysis.

The development of highly efficient electrochemical catalysts for the hydrogen evolution reaction (HER) through water splitting is a critical step in the advancement of hydrogen production for energy storage and conversion in modern industry[1–6]. Furthermore, the simple HER is a less sophisticated process in terms of understanding the mechanism of the water catalytic reaction and the relationship between the electrocatalytic activity and crystal structure at the nanoscale than the four-step oxygen evolution reaction/oxygen reduction reaction

processes. According to Trassati's volcano plot, rhodium (Rh) or Rh-based materials are promising catalysts for the HER[7–9]. However, the adsorption energy of hydrogen ($\Delta G_H$) on the Rh surface is still relatively high, which is unfavourable for the formation of H$_2$. In addition, the poor durability of these materials makes it necessary to design structures to achieve enhanced HER performance[9].

It is well known that nanosized entities in periodic identical arrays strongly influence the electronic and transport properties of the

[1]Institute of Functional Nano & Soft Materials (FUNSOM), Jiangsu Key Laboratory for Carbon-Based Functional Materials & Devices, Soochow University, Suzhou 215123 Jiangsu, China. [2]College of Chemistry, Chemical Engineering and Materials Science, Soochow University, Suzhou 215123 Jiangsu, China. [3]College of Energy, Soochow University, Suzhou 215123 Jiangsu, China. [4]Macao Institute of Materials Science and Engineering, Macau University of Science and Technology, Taipa 999078 Macau SAR, China. [5]Max Planck Institute for Chemical Physics of Solids, Nothnitzer Strasse 40, Dresden 01187, Germany. [6]These authors contributed equally: Zhenglong Fan, Fan Liao, Yujin Ji. ✉e-mail: yangl@suda.edu.cn; zhkang@suda.edu.cn; mwshao@suda.edu.cn; Zhiwei.Hu@cpfs.mpg.de; qshao@suda.edu.cn

material, providing collective characteristics different from those of the corresponding bulk structures[10,11]. To date, many periodic nanostructures have been reported, showing promise in applications in energy conversion, catalysis, and photoelectronic devices[12–17]. Notably, a perfectly aligned nanocrystal array with an interparticle distance of a few nanometres may provide a platform for pursuing different catalytic properties. However, the traditional nanolithography and template methods employed to fabricate the aligned array assembly always yield interparticle distances larger than 10 nm[18]. Thus, developing a strategy to fabricate a perfectly aligned nanocrystal array with a short interparticle distance (less than 5 nm) is highly desirable.

Two-dimensional (2D) metastable metal oxides may provide an ideal substrate for overcoming the above challenges. 2D materials have attracted extensive attention due to their maximum atomic utilization, ideal activities and desirable durability[19–32]. In addition, metastable metal oxides provide extensive possibilities for synthesizing the interfacial structures due to their intrinsic metastable properties[33–35]. Furthermore, in the metal/oxide catalytic interfacial system, the hydrogen spillover effect can cause the activated hydrogen atoms to migrate from a hydrogen-rich area to a hydrogen-poor area, which may provide an effective way to further improve the HER activity[36,37].

In this work, we report a perfectly aligned nanocrystal array on a pristine 2D metastable trigonal rhodium oxide (P-Tri-RhO$_2$). P-Tri-RhO$_2$ was fabricated by a radiofrequency-assisted molten-alkali method, and its crystal structure is categorized as space group P-3m1 (164), with lattice constants of $a = b = 3.091$ Å and $c = 4.407$ Å. More importantly, the 0.45% lattice mismatch between metastable P-Tri-RhO$_2$ and face-centred cubic (fcc) phase Rh leads to the in situ growth of Rh single-crystal nanoarrays with a short interparticle spacing of 3.709 nm. Such nanoscale spacing enables the spillover of hydrogen atoms to greatly enhance the HER with an ultralow overpotential of 9.8 mV at a current density of −10 mA cm$^{-2}$, a low Tafel slope of 24.0 mV dec$^{-1}$ and limited activity decay under a high potential (vs. RHE) of −0.4 V.

## Results

### Preparation and structure characterization of P-Tri-RhO$_2$

Pristine trigonal RhO$_2$ (P-Tri-RhO$_2$) was synthesized via a radiofrequency assisted molten-alkali method (Supplementary Fig. 1), where rhodium (III) chloride (RhCl$_3$) and potassium hydroxide (KOH) were selected as the raw materials. The final product of P-Tri-RhO$_2$ is brown (Supplementary Fig. 2a). Scanning electron microscopy (SEM) and transmission electron microscopy (TEM) were applied to characterize the morphology of P-Tri-RhO$_2$, revealing its 2D ultrathin nanosheet morphology (Fig. 1b, c and Supplementary Fig. 2b). The selected area electron diffraction (SAED) patterns of the P-Tri-RhO$_2$ sheet are hexagonal (Supplementary Fig. 2c), consistent with its trigonal structure. Atomic force microscopy (AFM) images reveal that the thickness of P-Tri-RhO$_2$ is approximately 1.39 nm (Fig. 1d). The crystal structure of P-Tri-RhO$_2$ is first revealed by X-ray diffraction (XRD), as shown in Fig. 1a. The crystal parameters are determined to be $a = b = 3.091$ Å and $c = 4.407$ Å. In addition, the simulated XRD pattern of P-Tri-RhO$_2$ is almost the same as the XRD pattern of P-Tri-RhO$_2$, further confirming its trigonal phase (Supplementary Fig. 3). Energy dispersive X-ray spectroscopy (EDX) results reflect that Rh and O are uniformly distributed in P-Tri-RhO$_2$ and that there is no K signal (Supplementary Fig. 4a–e). The elemental analysis results (elementar EL III) suggest that the atomic ratio of Rh and O is approximately 1:2 (Supplementary Fig. 4f). The Brunauer–Emmett–Teller (BET) surface area of P-Tri-RhO$_2$ (30.3 m$^2$ g$^{-1}$) is 3.19 times larger than that of Rutile-RhO$_2$ (9.5 m$^2$ g$^{-1}$) (Supplementary Fig. 5), indicating that P-Tri-RhO$_2$ nanosheets may possess more surface area. The crystal lattice of the P-Tri-RhO$_2$ sheet is clearly revealed in aberration-corrected dark-field scanning transmission electron microscopy (STEM-ADF) images (Fig. 1e–g), showing that

one Rh atom is surrounded by six adjacent Rh atoms with an intersection angle of 60°. As the intensity of atomic columns is proportional to the atomic number, a hexagonal pattern of Rh atoms is found, while the intensity of oxygen columns is too weak to be seen. The distance between two adjacent Rh atoms is determined to be 0.31 nm via STEM-ADF imaging, which is almost the same as the XRD result. P-Tri-RhO$_2$ completely transfers to rutile phase RhO$_2$ (Rutile-RhO$_2$) under annealing at 650 °C (Supplementary Fig. 6), reflecting its metastable nature.

In addition, the detailed electronic structures of P-Tri-RhO$_2$ were studied by X-ray photoelectron spectroscopy (XPS) and synchrotron-based X-ray absorption spectroscopy (XAS). Information on the electronic state of P-Tri-RhO$_2$ was first revealed by XPS (Supplementary Fig. 7). The peaks at 309.1 eV and 313.9 eV are completely attributed to Rh$^{4+}$ 3$d_{5/2}$ and Rh$^{4+}$ 3$d_{3/2}$ compared to those of Rutile-RhO$_2$[38], suggesting that a high purity phase of P-Tri-RhO$_2$ is obtained by the radiofrequency-assisted molten-alkali method. X-ray absorption near-edge structure (XANES) and extended X-ray absorption fine structure (EXAFS) analyses are highly sensitive to the electronic structure and the local environment of transmission metal ions[39–42]. Figure 1h shows the Rh-K XANES spectra of P-Tri-RhO$_2$ together with Rutile-RhO$_2$ and Rh foil for comparison. The absorption edge of P-Tri-RhO$_2$ is close to that of Rutile-RhO$_2$, indicating that the valence state of Rh in P-Tri-RhO$_2$ is close to +4. The Fourier transforms of P-Tri-RhO$_2$, Rh foil and Rutile-RhO$_2$[43] in Fig. 1i show the scattering profile as a function of the radial distance from the central absorbing Rh atom. The first peak at approximately 1.7 Å is assigned to Rh-O coordination, as found in Rutile-RhO$_2$[43] (blue line) and P-Tri-RhO$_2$ (black line), which have a second peak at approximately 2.8 Å related to the Rh-Rh shell. The spectrum for P-Tri-RhO$_2$ is fitted, as shown in Supplementary Fig. 8a, with the two coordination shells corresponding to Rh-O and Rh-Rh shells. The coordination number and bond lengths for these two shells are fitted to 6/2.02 ± 0.16 Å and 6/3.10 ± 0.01 Å, respectively, which is consistent with the XRD and STEM-ADF results. All these results allow us to conclude that a phase of RhO$_2$ with space group No. 164 (P-3m1) has been successfully prepared (Supplementary Table 1); the corresponding structure of P-Tri-RhO$_2$ is clearly shown in Fig. 1j–l.

In general, metastable phase materials usually require harsh synthetic conditions because they have higher Gibbs free energies than thermodynamically stable phase materials. In our synthetic process, high energy is mainly supplied via radiofrequency heating due to its rapid heating capabilities, which contributes to the formation of metastable phase materials. Only the amorphous product is obtained when directly RhCl$_3$ and KOH are directly mixed without the application of radiofrequency heating (Supplementary Fig. 9a, b). Rh$_2$O$_3$ is obtained when RhCl$_3$ is directly heated without the addition of KOH (Supplementary Fig. 9c, d). The above experiments indicate the important roles of high energy input and alkaline conditions in synthesizing metastable phase materials. In addition, to verify the chemical stability of metastable P-Tri-RhO$_2$, we performed more contrast experiments, as shown in Supplementary Fig. 10. The experimental results show that there are no morphology or crystal structure changes in P-Tri-RhO$_2$ after different treatments, indicating its excellent chemical stability.

### Preparation and structural characterization of Rh-NA/RhO$_2$

Well-ordered nanocrystal arrays (Rh-NA/RhO$_2$) were then prepared by electrochemically reducing P-Tri-RhO$_2$ by means of the chronoamperometry method at a constant reduction potential (vs. RHE) of −0.4 V for 2 h (Fig. 2a). The XRD pattern of Rh-NA/RhO$_2$ suggests that only a small amount of metallic Rh formed in Rh-NA/RhO$_2$ (Supplementary Fig. 11). In addition, the atomic ratio of metallic Rh and P-Tri-RhO$_2$ in the Rh-NA/RhO$_2$ electrocatalyst is approximately 1:4 according to the XPS results (Supplementary Fig. 12). Next, the electrochemical specific surface area (ECSAs) of Rh-NA/RhO$_2$ was determined from

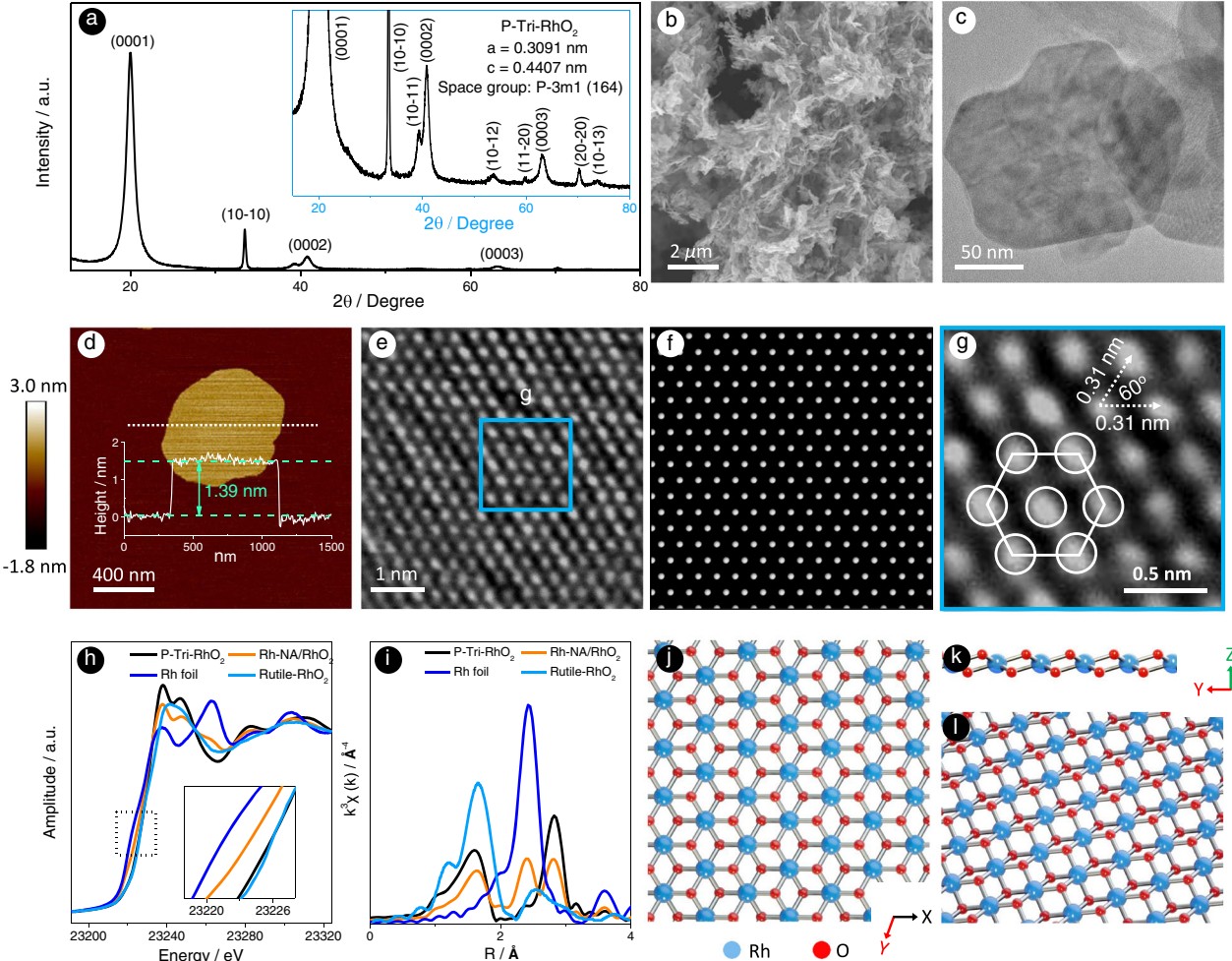

**Fig. 1 | Structure characterization of P-Tri-RhO₂ and Rh-NA/RhO₂. a** XRD pattern and its enlarged view. **b** The SEM and (**c**) TEM images, showing its 2D nanosheets morphology. **d** AFM image and the corresponding height profile. **e** STEM image, clearly showing the atomic arrangement of Rh. **f** Simulated STEM image and (**g**) partial enlargement from (**e**). **h** Normalized Rh K-edge XANES spectra for P-Tri-RhO₂, Rh-NA/RhO₂, Rh foil and standard Rutile-RhO₂. **i** Normalized Fourier transformed (FT) k³-weighted χ (k)-function of the extended X-ray absorption fine structure (EXAFS) spectra for P-Tri-RhO₂, Rh-NA/RhO₂, Rh foil and standard Rutile-RhO₂ reference at Rh K-edge. **j–l** The corresponding atomic models of P-Tri-RhO₂ from different directions.

hydrogen under potential deposition (ECSA$_{Hupd}$) by obtaining cyclic voltammograms (CVs) in 0.5 M H₂SO₄ with a scan rate of 50 mV s⁻¹. As shown in Supplementary Fig. 13 and Supplementary Table 2, the ECSA$_{Hupd}$ of P-Tri-RhO₂ increases from 8.8 to 55.5 m² g$_{Rh}$⁻¹ after the 2 h electroreduction test, further suggesting the in situ formation of Rh nanoparticles on the P-Tri-RhO₂ substrate.

To further demonstrate the structure of Rh-NA/RhO₂, scanning transmission electron microscopy (STEM) images of Rh-NA/RhO₂ are shown in Fig. 2b–d, which clearly reveal that the Rh nanocrystal array is uniformly distributed in a hexagonal pattern on the substrate of P-Tri-RhO₂. A STEM image at high magnification shows that the Rh particles are single crystalline in nature with a diameter of ~3.2 nm. More importantly, the interparticle distance of this array is measured to be 3.709 nm, which means there is a molecular-scale distance (~0.5 nm) between two adjacent particles (Fig. 2c). As a comparison, we conducted the same in situ electrochemical reduction step for Rutile-RhO₂, and the corresponding results are shown in Supplementary Fig. 14. The TEM and HRTEM images of the above product indicate that the Rh nanoparticles can be reduced in situ on Rutile-RhO₂, while no Rh nanocrystal arrays are observed, suggesting the key role of the metastable two-dimensional P-Tri-RhO₂ precursor in forming this special array structure (Supplementary Fig. 14b–d).

It is worth noting that these Rh single-crystal particles are well ordered, so all the crystal lattices of the Rh {220} planes are shown in the same orientation (Fig. 2e). The (20-2), (2-20) and (02-2) planes are marked in cyan, yellow and white, respectively. Such particles aligned in an arrayed manner may originate from the intrinsic anisotropy of metastable RhO₂, which compels Rh particles to grow preferentially along certain crystallographic directions.

To better explain the growth mechanism for the single crystal-line Rh array, the mismatch between the (20-20) plane of P-Tri-RhO₂ and the (220) plane of face-centred cubic Rh is calculated, as shown in Supplementary Fig. 15. The cell parameters of *a* for P-Tri-RhO₂ and face-centred cubic Rh are 0.3091 and 0.3803 nm, respectively. The values of the *d*-spacing of the P-Tri-RhO₂ (20-20) plane and Rh (220) plane are determined to be 0.1339 and 0.1345 nm, respectively, indicating that there is only 0.45% ((0.1345−0.1339) × 100% / 0.1339 = 0.45%) mismatch. Such a small mismatch makes the in situ epitaxial growth of face-centred cubic Rh on the P-Tri-RhO₂ substrate feasible. Further magnification of the STEM images of Rh-NA/RhO₂ clearly proves that face-centred cubic Rh with a three-layer structure is generated in situ on the substrate of P-Tri-RhO₂ by epitaxial growth along the [111] direction of Rh (Fig. 2f, g).

While these nanocrystal arrays are similar to Moire patterns at first glance, careful comparison shows that they are totally different. We

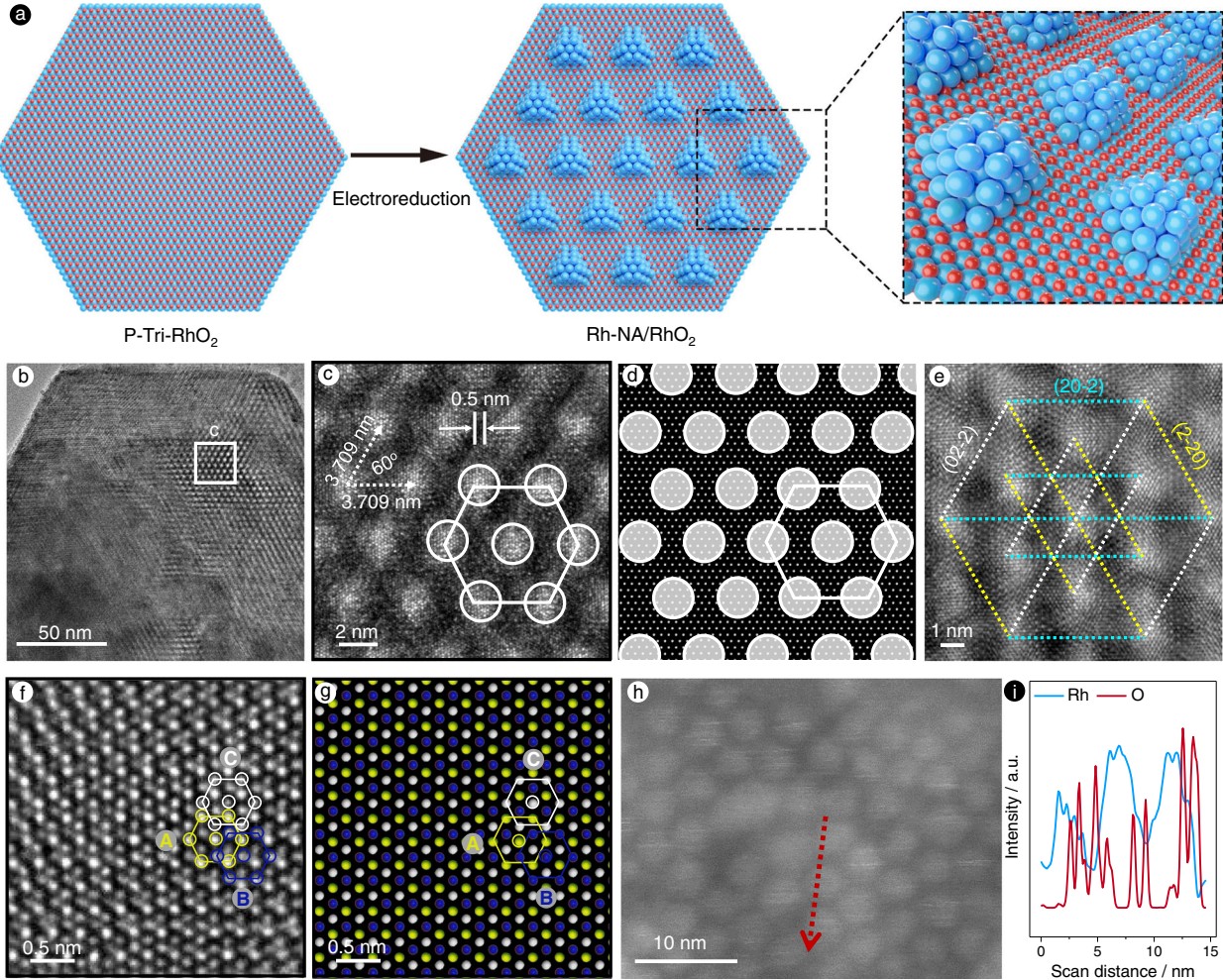

**Fig. 2 | Structure characterization of Rh-NA/RhO₂. a** Schematic representation for in-situ growth of molecular-scale-spacing, well ordered and single crystalline Rh array on substrate of P-Tri-RhO₂ to obtain Rh-NA/RhO₂ electrocatalyst. **b** The STEM image of Rh-NA/RhO₂ and (**c**) its partial enlargement, clearly showing single crystalline Rh atom array on the surface of P-Tri-RhO₂. **d** Simulated pattern for (**c**). **e** The enlarged STEM images of Rh-NA/RhO₂, clearly showing {220} crystal lattices of Rh particles with well ordered arrangement: (20-2), (2-20) and (02-2) planes were marked with the cyan, yellow and white colors respectively. **f** The further magnification for STEM image of Rh-NA/RhO₂. **g** Simulated STEM image for (**f**), clearly showing three layers structure of face-centered cubic Rh. The Rh atoms on different layers are marked with green, red and white colours. **h** The HAADF-STEM image and (**i**) the corresponding EDX line scanning profile of Rh-NA/RhO₂. Figure 2a was made with the Cinema 4D Software.

first simulated Moire patterns by twisting the bilayer P-Tri-RhO₂ region from 0 to 30° with increments of 1°, as shown in Supplementary Movie 1. These Moire patterns all have a similar arrangement. Taking the Moire pattern with a rotation angle of 3° as an example, a small region of the simulated Moire pattern (Supplementary Fig. 16a) is similar to the real experimental HRTEM image (Fig. 2c), but in larger regions, the Moire patterns (Supplementary Fig. 16b, c) are different from the real experimental and simulated patterns (Fig. 2f, g). Moreover, we compared the surface densities (SDs, the number of Rh atoms per nm²) of Rh atoms of Rh-NA/RhO₂ and different Moire patterns. The theoretical maximum SDs of Rh atoms in the Moire patterns constructed by two single layers of P-Tri-RhO₂, two single layers of metallic Rh, or a single layer of P-Tri-RhO₂ and a single layer of metallic Rh are determined to be 24.172, 31.938 or 28.055 nm⁻², respectively (Supplementary Note 1 and Supplementary Table 3), much lower than those of Rh nanocrystal arrays in Rh-NA/RhO₂ (the theoretical and actual SDs are 47.904 and 47.3 ± 1.2 nm⁻², respectively). We also simulated a Moire pattern by rotating a single layer of P-Tri-RhO₂ and a single layer of metallic Rh. As shown in Supplementary Fig. 17, a typical Moire pattern is obtained by twisting a single layer of P-Tri-RhO₂ and a single layer of

metallic Rh with a rotation angle of 3°. The atomic enlarged areas of this Moire pattern are completely different from those of the Rh nanocrystal array (Fig. 2f, g), indicating that the Rh nanocrystal array is real rather than a Moire pattern. The Rh nanocrystal arrays are clearly observed in the HAADF-TEM image (Fig. 2h). The EDX line scanning profile in Fig. 2i shows that the contrasts on the Rh and O elements demonstrate nearly equal spaced Rh nanocrystals in a consistent way, which also excludes the formation of Moire pattern. Furthermore, we also simulated the XRD patterns of different atomic layers of Rh. As shown in Supplementary Fig. 18, the simulated XRD peaks of different atomic layers of Rh cannot be detected in Rh-NA/RhO₂, excluding a thin Rh atomic layer on P-Tri-RhO₂.

The XPS spectrum of Rh-NA/RhO₂ is also shown in Supplementary Fig. 12, where the peaks at 307.3 eV and 312.1 eV are attributed to metallic Rh 3d$_{5/2}$ and 3d$_{3/2}$, respectively[44]. Compared with C-Rh/C, the binding energy of Rh in Rh-NA/RhO₂ shifts to a higher binding energy by approximately 0.15 eV, indicating the existence of a strong electronic interaction between the Rh nanocrystal array and the P-Tri-RhO₂ substrate[43]. As shown in Fig. 1h, the energetic position of Rh-NA/RhO₂ is located between those of Rutile-RhO₂ and the Rh foil, indicating that the Rh ions in Rh-NA/RhO₂ are reduced[45,46]. As shown

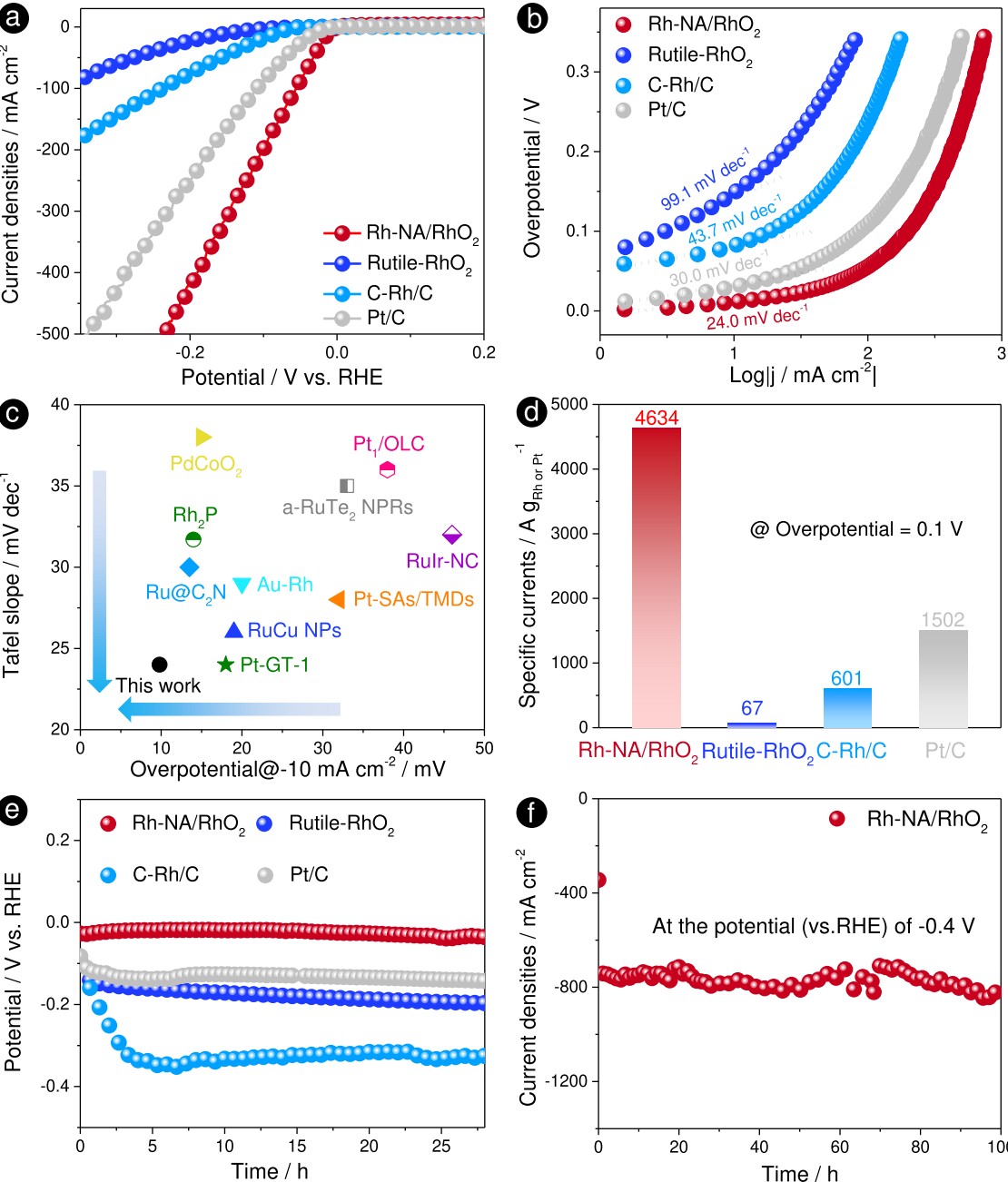

**Fig. 3 | HER performances of Rh-NA/RhO$_2$, Rutile-RhO$_2$, C-Rh/C and Pt/C in H$_2$-saturated 0.5 M H$_2$SO$_4$ electrolyte. a** The HER polarization curves of Rh-NA/RhO$_2$, Rutile-RhO$_2$, C-Rh/C and Pt/C with $i$R-correction. **b** Tafel plots obtained from the polarization curves of Rh-NA/RhO$_2$, Rutile-RhO$_2$, C-Rh/C and Pt/C in Fig. 3a. **c** Comparison of Tafel slopes and overpotentials at the current density of −10 mA cm$^{-2}$ for Rh-NA/RhO$_2$ with previous reported high activity noble metal based HER catalysts (Pt-GT-1[59], RuCu NPs[60], Pt-SAs/TMDs[61], Au-Rh[62], Ru@C$_2$N[63], Rh$_2$P[64], RuIr-NC[65], a-RuTe$_2$ NPRs[66], PdCoO$_2$[67] and Pt$_1$/OLC[68]). **d** Comparison of specific currents for Rh-NA/RhO$_2$, Rutile-RhO$_2$, C-Rh/C and Pt/C electrocatalysts at the overpotential of 0.1 V. **e** Stability of Rh-NA/RhO$_2$, Rutile-RhO$_2$, C-Rh/C and Pt/C electrocatalysts by the chronopotentiometry technique at a constant current density of −10 mA cm$^{-2}$. **f** The stability test for Rh-NA/RhO$_2$ by chronoamperometry test under the high potential (vs. RHE) of −0.4 V for 100 h.

in Fig. 1i, for Rh-NA/RhO$_2$, a peak occurs at 2.2 Å, which is located at the same position as that in Rh foil (green line), again indicating the reduction of Rh, in agreement with the Rh-K XANES results. The spectrum for Rh-NA/RhO$_2$ is fitted and shown in Supplementary Fig. 8d with three obvious coordination shells corresponding to two shells for RhO$_2$ and one for the reduced Rh. The coordination number and bonding lengths for these three shells are fitted to 6/3.10 ± 0.01 Å, 12/2.68 ± 0.08 Å and 6/2.02 ± 0.16 Å, respectively, clearly indicating that face-centred cubic Rh is generated in situ on the P-Tri-RhO$_2$ substrate after electroreduction.

## Electrochemical performance of Rh-NA/RhO$_2$

In the following study, we evaluated the HER catalytic activity of Rh-NA/RhO$_2$ in the H$_2$-saturated 0.5 M H$_2$SO$_4$ via a three-electrode system. Before electrochemical tests, two Pt electrodes were used as the working and counter electrodes to calibrate the SCE (Supplementary Fig. 19). The HER performances of Rh-NA/RhO$_2$, Rutile-RhO$_2$, C-Rh/C and Pt/C are shown in Fig. 3a, where Rh-NA/RhO$_2$ exhibits better HER activity than Rutile-RhO$_2$, C-Rh/C and Pt/C. The overpotentials at a current density of −10 mA cm$^{-2}$ are summarized in Supplementary Fig. 20 and Supplementary Table 4, with which the HER activities can

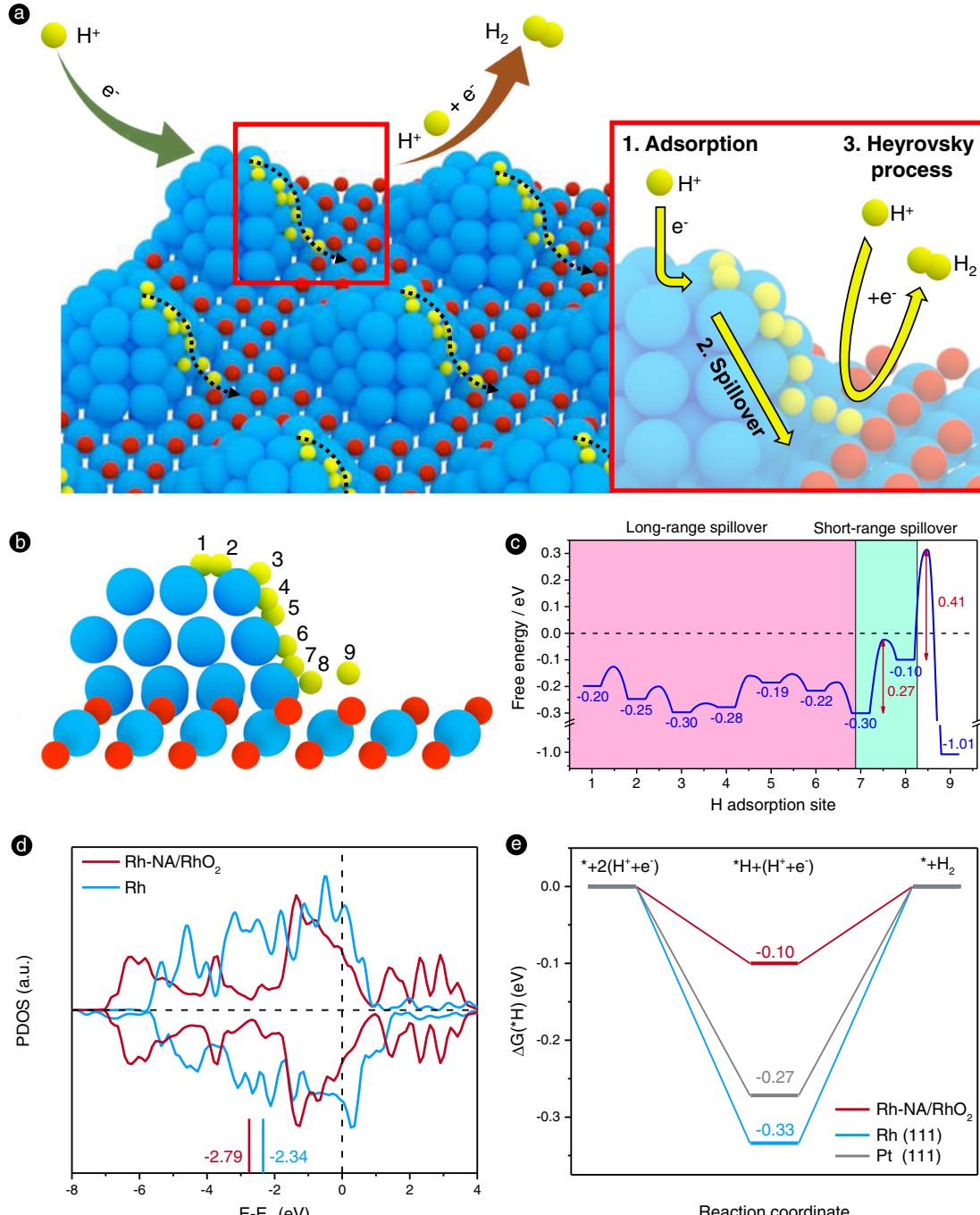

**Fig. 4 | Schematic of hydrogen spillover assisted HER mechanism and DFT calculation of Gibbs free energy evolution. a** The schematic representation of hydrogen spillover assisted HER mechanism for Rh-NA/RhO₂ electrocatalyst. The enlarge image shows that the HER process in Rh-NA/RhO₂ system includes three main steps: (i) adsorption and reduction of hydrogen on the Rh nanoparticles surface; (ii) the hydrogen spillover from Rh nanoparticles to the contact interface between Rh nanoparticles and P-Tri-RhO₂ substrate; and (iii) the Heyrovsky process to produce hydrogen molecules. **b** The hydrogen spillover effect on Rh-NA/RhO₂. **c** The hydrogen adsorption free energy diagram of different sites on Rh-NA/RhO₂. **d** Comparisons of the *d*-orbitals distribution of Rh and Rh-NA/RhO₂. **e** Two-electron Gibbs free energy evolution diagrams on Pt (111), Rh (111) and Rh-NA/RhO₂.

be quantitatively compared. In detail, Rh-NA/RhO₂ delivers a low overpotential of 9.8 mV at a current density of −10 mA cm⁻², which is much lower than those of Rutile-RhO₂ (148 mV), C-Rh/C (83 mV) and Pt/C (29 mV). The corresponding Tafel slopes of Rh-NA/RhO₂, Rutile-RhO₂, C-Rh/C and Pt/C are 24.0, 99.1, 43.7 and 30.0 mV dec⁻¹, respectively (Fig. 3b), indicating that Rh-NA/RhO₂ exhibits the fastest kinetic rate towards hydrogen evolution.

The exchange current density ($|j_0|$) of Rh-NA/RhO₂ is 3.356 mA cm⁻², 2.75 times higher than that of Pt/C (1.220 mA cm⁻²), indicating its excellent HER electrocatalytic activity (Supplementary Table 5). Then, the specific current of Rh-NA/RhO₂ at −0.1 V (vs. RHE) is up to 4634 A g$_{Rh}^{-1}$, 69.1, 7.7 and 3.1 times higher than those of Rutile-RhO₂ (67 A g$_{Rh}^{-1}$), C-Rh/C (601 A g$_{Rh}^{-1}$) and Pt/C (1502 A g$_{Pt}^{-1}$), respectively (Fig. 3d and Supplementary Fig. 21). The electrochemical specific surface form hydrogen under potential deposition (ECSA$_{Hupd}$) of Rh-NA/RhO₂, Rutile-RhO₂, C-Rh/C and Pt/C are determined to be 55.5 m² g$_{Rh}^{-1}$, 8.3 m² g$_{Rh}^{-1}$, 15.3 m² g$_{Rh}^{-1}$ and 47.3 m² g$_{Pt}^{-1}$ respectively (Supplementary Fig. 22 and Supplementary Table 2). Next, the stability of Rh-NA/RhO₂, Rutile-RhO₂,

C-Rh/C and Pt/C was evaluated by chronopotentiometry at a constant current density of −10 mA cm⁻². In addition, the overpotential (vs. RHE) at −10 mA cm⁻² and Tafel slope of Rh-NA/RhO₂ are also compared with those of previously reported high-activity noble metal-based HER catalysts (Fig. 3c and Supplementary Table 6), showing that Rh-NA/RhO₂ is among the best electrocatalysts for hydrogen evolution. To further eliminate the effect of particle size and surface area, the turnover frequencies (TOFs) were calculated to compare the catalytic activities of different catalysts. As shown in Supplementary Fig. 23a, Rh-NA/RhO₂ always has a higher TOF than Rutile-RhO₂, C-Rh/C and Pt/C at different potentials (vs. RHE). In detail, a TOF of 4.68 s⁻¹ is achieved with Rh-NA/RhO₂ at overpotentials of 20 mV, 17.3, 18 and 6.5 times higher than those of Rutile-RhO₂ (0.27 s⁻¹), C-Rh/C (0.26 s⁻¹) and Pt/C (0.72 s⁻¹) (Supplementary Fig. 23b), indicating the excellent HER intrinsic activity of Rh-NA/RhO₂. As shown in Fig. 3e, the potential (vs. RHE) of Rh-NA/RhO₂ shifts only 7 mV after the 28 h stability test. In a sharp comparison, the Rutile-RhO₂, C-Rh/C and Pt/C drop 99, 166 and 60 mV after 28 h, respectively, suggesting that Rh-NA/RhO₂ exhibits not only excellent HER activity but also long-term durability.

To reveal the structural change in the Rh nanocrystal array after the harsh long-term hydrogen evolution reaction, a chronoamperometry test was carried out under a high potential (vs. RHE) of −0.4 V. As shown in Fig. 3f, Rh-NA/RhO₂ exhibits superior stability over a 100 h long-term test, with the average current density reaching −750 mA cm⁻². The XRD pattern shows that there are no obvious crystal structure changes in Rh and Tri-RhO₂ after the stability test (Supplementary Fig. 24a). SEM and STEM images of Rh-NA/RhO₂ after the stability test indicate that the structural change in the Rh nanocrystal array is very limited (Supplementary Fig. 24b, c). XPS (C1s, O1s and F1s) and FTIR spectra of Rh-NA/RhO₂ were collected after the long-term HER stability test and are shown in Supplementary Fig. 25. The C-F, CH₃ and O-H bonds can be clearly observed from FTIR and XPS spectra, proving the existence of Nafion and isopropanol in the Rh-NA/RhO₂ electrocatalyst[47,48].

## The hydrogen spillover of Rh-NA/RhO₂

The low overpotential and small Tafel slope of Rh-NA/RhO₂ might be attributed to the unique interaction between the nanocrystal hexagonal array and the two-dimensional metastable substrate. The molecular-scale distance between two adjacent Rh particles is only 0.5 nm, favouring hydrogen spillover[49–51]. The functional properties of arrays heavily depend on their density, their interparticle distance, their orientation and the uniformity of the particles, as well as the crystal quality[52]. The arrays with large interparticle spacing maintain their individual properties, while those short interparticle distance exhibit strong coupling between particles[16]. As listed in Supplementary Table 7, R Rh-NA/RhO₂ has the shortest interparticle distance, leading to strong coupling between adjacent particles in HER catalysis and thus exhibiting enhanced HER catalytic performance.

A hydrogen spillover-assisted HER mechanism for the Rh-NA/RhO₂ catalyst was proposed, and a detailed process is shown in Supplementary Note 2. To confirm the hydrogen spillover phenomenon for the Rh-NA/RhO₂ electrocatalyst, the physical mixtures of Rh-NA/RhO₂ and WO₃ were treated under a H₂ atmosphere at room temperature (Supplementary Fig. 26b). As expected, the yellow WO₃ particles turn dark blue[51], indicating the existence of a hydrogen spillover effect in the Rh-NA/RhO₂ electrocatalyst. We also performed the same experiments by using the physical mixtures of P-Tri-RhO₂ and WO₃ (P-Tri-RhO₂-WO₃) and Rh and WO₃ (Rh-WO₃). As shown in Supplementary Fig. 26c, no colour change is observed in P-Tri-RhO₂-WO₃, suggesting no hydrogen spillover generation. As shown in Supplementary Fig. 26d, the physical mixture of Rh-WO₃ turns to a dark-blue colour after H₂ treatment, indicating the existence of hydrogen spillover in metallic Rh.

We then performed experiments to reveal the active sites of Rh-NA/RhO₂ for the hydrogen evolution process by adding thiocyanate (SCN⁻) or tetramethylammonium cation (TMA⁺) to an acidic electrolyte since SCN⁻ and TMA⁺ have specific interactions with metal and negative oxygenated species, respectively[53]. As shown in Supplementary Fig. 27, Rh-NA/RhO₂ exhibits obvious performance decay after the addition of these two chemical probes, indicating that both Rh and RhO₂ play important roles in the HER. However, the metallic Rh catalyst shows obvious performance loss after the addition of SCN⁻ and has nearly no performance decay after the addition of TMA⁺, showing that metallic Rh is the sole active site and that TMA⁺ has no effect on Rh.

## Theoretical simulation of the HER on Rh-NA/RhO₂

Density functional theory (DFT) calculations were further employed to explore the underlying HER enhancement mechanism behind the unique interface structure of Rh-NA/RhO₂. A Rh nanoparticle was placed on the basal plane of the (6 × 6) supercells of the 2D P-Tri-RhO₂ nanosheet to represent the structural model of Rh-NA/RhO₂. We calculated the hydrogen adsorption free energies ΔG(*H) on Rh-NA/RhO₂ to investigate the potential HER active sites, including the Rh nanoparticle, the contact interface between the Rh nanoparticle and the P-Tri-RhO₂ substrate, and the basal plane of the P-Tri-RhO₂ substrate (Fig. 4a–c). It is found that the basal O of Tri-RhO₂ (site 9) has a strong hydrogen affinity with a ΔG(*H) of −1.01 eV, making it hard for the *H to diffuse due to its high energy barrier, 1.32 eV. By comparison, the *H on Rh nanoparticles (sites 1-7) are weaker, with ΔG(*H) of −0.30–−0.19 eV. It should be noted that the ΔG(*H) on the contact interface (site 8) is −0.10 eV, which is closer to zero for an ideal HER activity favourable for H₂ formation when combined with a free proton in solvation. Moreover, the largest energy barrier of hydrogen spillover is only 0.27 eV from site 7 to 8. Therefore, our DFT calculations confirm the long-range and short-range hydrogen migration from the Rh nanoparticle to the contact interface (from site 1–7 to site 8) for H₂ formation. P-Tri-RhO₂ acts as a charge collector, attracting electrons from Rh nanoparticles. Therefore, the d-band centre of Rh-NA/RhO₂ decreases by −2.79 eV, which is lower than that of Rh (111) (−2.34 eV) (Fig. 4d and Supplementary Fig. 28). The decrease in the d-band centre is helpful for weakening the surface hydrogen adsorption and modulating ΔG(*H) from −0.33 eV on Rh(111) to −0.10 eV on Rh-NA/RhO₂, closer to zero than the −0.27 eV on Pt(111) (Fig. 4e). This also explains why our synthesized Rh-NA/RhO₂ exhibits better HER activity than Pt-based electrocatalysts in the experiments. Therefore, we ultimately can reveal the hydrogen spillover-enhanced HER process in the Rh-NA/RhO₂ system (Fig. 4a): (i) adsorption and reduction of hydrogen on Rh nanoparticles; (ii) hydrogen spillover from Rh nanoparticles to the contact interface; and (iii) the Heyrovsky process to produce hydrogen molecules on the contact interface.

## Discussion

In conclusion, a metastable oxide, trigonal phase RhO₂, was successfully obtained via a radiofrequency heating method. Using trigonal phase RhO₂ as the precursor, a well-ordered nanocrystal array on trigonal rhodium oxide was formed with a limited intersurface distance of 0.5 nm. The unique coupling between the nanocrystal array and the 2D metastable substrate enables effective hydrogen spillover, enhancing the hydrogen evolution reaction, with an ultralow Tafel slope of 24.0 mV dec⁻¹ and an overpotential of only 9.8 mV at a current density of −10 mA cm⁻². This work yields an important method for fabricating a well-ordered nanocrystal array with subnanometre spacing for future advanced applications.

## Methods
### Chemicals
Rhodium (III) chloride (RhCl₃) was purchased from Aladdin Industrial Co. 10% Rh/C was purchased from Shanghai Haohong Biomedical

Technology Co. 20% Pt/C was obtained from Aladdin Industrial Co. Potassium hydroxide and potassium thiocyanate (KOH, 99%) were purchased from Sinopharm Chemical Reagent Co. Hydrochloric acid (HCl, Guaranteed reagent) was purchased by Chinasun Specialty Products Co. Nafion solution (5 wt%) and Tetramethylammonium chloride were supported by Sigma–Alddrich Co. Isopropanol (99.8%) was obtained from Sinopharm Chemical Reagent Co. Toray carbon paper (TGP-H-60) was bought from Alfa Aesar. Other reagents were of analytical reagent grade without further purification. Double-distilled water was used all experiments.

## Synthesis of P-Tri-RhO₂

The pristine trigonal $RhO_2$ (P-Tri-$RhO_2$) was obtained via a radio-frequency heating method. Specially, 300 mg $RhCl_3$ and 10 g KOH were mixed in a high-quality corundum crucible. Then the radio frequency heater with the power of 15 kW was used to heat above mixture for 20 mins, and then naturally cooled to room temperature. The obtained product was washed by 1 M HCl and redistilled water for several times respectively, dried in air for 6 h to obtain P-Tri-$RhO_2$. Rutile-$RhO_2$ was obtained by annealing P-Tri-$RhO_2$ at 650 °C in air for 2 h.

## Synthesis of Rh-NA/RhO₂

Rh nanocrystal array (Rh-NA/$RhO_2$) was obtained by in-situ growth of face-centered cubic Rh nanoparticles on the surface of P-Tri-$RhO_2$ via the electroreduction method. In brief, 20 mg P-Tri-$RhO_2$ was added into the mixed solution (9 mL isopropanol and 1 mL 0.5 wt% Nafion solution) and ultrasonicated to form the homogenous ink. 2 mL above dispersion was dropped on the surface of carbon paper (1 cm × 2 cm) and dried naturally. Then P-Tri-$RhO_2$ was reduced by conducting chronoamperometry method at a constant potential (vs. RHE) of −0.4 V for 2 h. After that, the above product was collected from carbon paper, cleaned by ethanol for several times and dried in air for 6 h to obtain the final product of Rh-NA/$RhO_2$.

## Structure characterization

X-ray powder diffraction (XRD, Philips X'pert PRO MPD diffractometer) with Cu Kα radiation source ($\lambda_{Cu} = 0.15406$ nm) was applied to study the phase and crystallography of all samples. The transmission electron microscopy (TEM) images and energy dispersive X-ray spectroscopy (EDX) of all samples were characterized via a FEI Tecnai F20 transmission electron microscope with an accelerating voltage of 200 kV. The content of O element in P-Tri-$RhO_2$ was determined by elemental analysis method (elementar EL III). Scanning transmission electron microscopy (STEM) results were collected on a fifth order aberration-corrected transmission electron microscope (JEOL ARM200CF) at 80 kV. The images have been filtered using a Gaussian filter to improve the contrast. Samples were baked at 140 °C for 8 h before taking into the microscope. Scanning electron microscopy (SEM) was performed by using a Zeiss G500. The chemical states of products were analyzed by using X-ray photoelectron spectroscopy (XPS) on a Kratos AXIS UltraDLD ultrahigh vacuum surface analysis system with Al Kα radiation (1486 eV) as a probe. The surface topographic height of P-Tri-$RhO_2$ was measured via the atomic force microscopy (AFM, Bruker Dimension Icon). The BET specific surface areas were characterized by American Micromeritics ASAP-2020 porosimeter. XAS data were collected at the SPring-8 BL14B2 (Harima Science Garden City, Hyogo) using a Si (111) quick-scanning monochromator with the transmission mode.

## Electrochemical measurements

CHI 760D electrochemical workstation with a standard three-electrode system was used to perform the HER experiments. A modified glassy carbon electrode (GCE, 3 mm in diameter or 0.0707 cm⁻² in area) and a saturated calomel electrode (SCE) were chosen as the working electrode and the reference electrode, respectively. A carbon rod was selected as the counter electrode. The catalysts solution was prepared as follows: 1 mg catalyst (Rh-NA/$RhO_2$ or Rutile-$RhO_2$) and 4 mg carbon black were added into the mixed solution (900 μL isopropanol and 100 μL 0.5 wt% Nafion solution) and ultrasonicated to form the homogenous ink. 4 μL (20 μg catalyst or 3.1 μg Rh) dispersion was dropped on the surface of GCE (43.8 μg cm$_{Rh}^{-2}$) and dried naturally for testing. In this work, isopropanol is served as a dispersant agent. Nafion is served as dispersand and binding ones and affects the proton transfer. Both of them are important for the dispersity, stability and proton conduction of the electrocatalysts during the HER process. In addition, 5 mg 20% Pt/C or 10% Rh/C electrocatalysts were prepared as same as above method. Then 4 μL (4 μg Pt or 2 μg Rh) dispersion was dropped on the surface of GCE (56.5 μg cm$_{Pt}^{-2}$ or 28.3 μg cm$_{Rh}^{-2}$) and dried naturally for testing. The HER performances were analyzed in H₂-saturated 0.5 M H₂SO₄ by linear sweep voltammetry (LSV) in the range of −0.4 to 0.2 V (vs. RHE) with the scan rate of 5 mV s⁻¹ and 95% iR-compensation. The electrochemical specific surface areas (ECSAs) of different catalysts determined from hydrogen under potential deposition by performing CV in 0.5 M H₂SO₄ with a scan rate of 50 mV s⁻¹ and ranging from 0 to 1.2 V (vs. RHE). The stability tests of Rh-NA/$RhO_2$, Rutile-$RhO_2$, C-Rh/C and Pt/C electrocatalysts by the chronopotentiometry technique were carried out at a constant current density of −10 mA cm⁻². The stability test for Rh-NA/$RhO_2$ by chronoamperometry test was carried out under the high potential (vs. RHE) of −0.4 V for 100 h. All electrochemical tests were carried out in ambient condition.

The corresponding equations are shown as follow:

Turnover frequencies (TOFs) of electrocatalysts were calculated as follows:

$$TOF = \frac{3.12 \times 10^{15} \frac{H_{2/s}}{cm^2} \, per \, \frac{mA}{cm^2} \times |j|}{active \, sites} \tag{1}$$

The active sites of electrocatalysts was calculated as follows:

$$Active \, sites = \frac{Q}{1.602 \times 10^{-19}} \tag{2}$$

$$Q = \frac{S_{peak}}{v} \tag{3}$$

Where $S_{peak}$ is the integral area of adsorbed hydrogen desorption peak in the CV curve (Supplementary Fig. 22), v is the scan rate of 50 mV s⁻¹ and Q is quantity of electric charge.

## Details of theoretical simulation

All theoretical simulations were under the framework of density functional theory with spin-polarized plane wave basis sets and implemented in the Vienna ab-initio Software Package (VASP) version 5.4.1[54,55]. The electronic exchange-correlation energy was described by the Perdew-Burke-Ernzerhof functional formula with consideration of the DFT-D3 correction method[56,57]. As the evaluation of electronic energies during self-consistent calculations, the cutoff energy was set to 450 eV and the convergence thresholds of energy and force were corresponding to 1E⁻⁴ eV and −0.05 eV Å⁻¹, respectively. The k-point sampling only adopted Gamma point, which is enough to simulate this large metal-support model. The hydrogen adsorption free energy was calculated by

$$\Delta G(*H) = \Delta E + 0.29 = E(*H) - E(*) - E(H_2) + 0.29 \tag{4}$$

in which the E(*H) and E(*) are the total energies of surface models with and without hydrogen adsorption, and the E(H₂) is the total energy of hydrogen molecule. The constant of 0.29 is regarded as the contribution of vibrational and entropic correction[58].

## Data availability

The data generated in this study are provided in the Supplementary Information/Source Data file. Source data are provided with this paper.

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

## Acknowledgements

This work was financially supported by National MCF Energy R&D Program (2018YFE0306105), Development Program of China (2017YFA0204800), National Key R&D Program of China (2020YFA0406104, 2020YFA0406101), National Natural Science Foundation of China (51725204, 21771132, 51972216, 52041202, 21771134, 51902217, 21905188), Natural Science Foundation of Jiangsu Province (BK20190041), Innovative Research Group Project of the National Natural Science Foundation of China (51821002), the major project of Basic Science (natural science) of Jiangsu Province (21KJA430001), Jiangsu Provincial Natural Science Foundation (BK20211316), the Suzhou Municipal Science and Technology Bureau (SYG202125), State Key Laboratory of Physical Chemistry of Solid Surfaces, Xiamen University (202113), the Collaborative Innovation Center of Suzhou Nano Science & Technology, the Priority Academic Program Development of Jiangsu Higher Education Institutions (PAPD), the 111 Project, Suzhou Key Laboratory of Functional Nano & Soft Materials and Joint International Research Laboratory of Carbon-Based Functional Materials and Devices.

## Author contributions

Q.S. conceived and supervised the research. Z.F., F.L., Q.S., W.Z., Y. Liu, Z.K., M.S., H.H., M.M., M.W., H.Y. and K.Y. performed most of the experiments and data analysis. D.W. conducted the STEM experiment. Z.H. analyses the EXAFS results. Y.J. and Y. Li performed and analyzed the DFT simulations. Z.F., Y. Liu, Z.K., M.S., Z.H. and Q.S. wrote the paper. All authors discussed the results and commented on the manuscript. Z.F., F.L. and Y.J. equally contribute to this work.

## Funding

## Competing interests

The authors declare no competing interests.
