## [Peer Review File · Nature Communications]

Coupling of Nanocrystal Hexagonal Array and Two-Dimensional Metastable Substrate Boosts H₂-ProductionREVIEWER COMMENTS

Reviewer #1 (Remarks to the Author):

This study describes the synthesis and fine characterization of a heterostructure comprising a metastable RhO₂ phase on which metallic rhodium particles have been formed. The proposed electrochemical method offers fine control of rhodium particle size and interparticle distance. The formed interface also enables H-spillover. All of these features combined result in excellent HER performance.

The proposed results are of interest to a broad community and may be published in this journal once the following changes have been made:

- 1) the introduction of the article must be completely rewritten. The ins and outs do not appear clearly and the scientific approach followed is very confused. The English in the introduction also needs to be reviewed.
- 2) a surface characterization of the Rh-NA/RhO₂ must be given. In particular the preparation of this material involves the use of an alcohol and nafion. The presence of these organic species on the surface of the material must be probed (XPS C1s, O1s, F1s; FTIR). The possible presence of such species on the reaction should, if confirmed, be discussed.

Reviewer #2 (Remarks to the Author):

Processes at electrochemical interfaces are complex. It is thus imperative to stress, that the electrochemical interface is the place where the formation of surface chemical complexes formation can be induced by the processes of adsorption, and charge transfer (usually multi-electronic). In this context, the hydrogen evolution reaction (HER) is a process of paramount importance, via water splitting, to generate H₂, a clean fuel. The rate determining step (rds) of complex electrochemical processes are changeable with different experimental conditions. Even at different intensity of polarizations, the rds of HER are different as the observed Tafel slopes of ~30 mV/decade at weak polarization and of ~120 mV at strong polarization. In a whole, electrocatalysis is very challenging in mechanisms and kinetics.

The present manuscript by Fan et al, illustrates the important role that a metastable material can play for the design of advanced materials. This can be categorized as a noteworthy result. The strategy requires the fabrication of well-ordered nanocrystal matrix with sub-nanometer spacing. Therefore, it is evident that the synthesis of electrocatalytic materials with molecular-scale distance and well-aligned structure could be of great utility for the development of a catalytic system for hydrogen evolution. In this regard, the authors discuss several possibilities by showing results on an array of well-ordered nanocrystals in a pristine 2D metastable rhodium oxide (P-Tri-RhO₂). Apparently, the unique interaction between the nanocrystal array and the 2D metastable substrate allows the spillover of hydrogen atoms, as discussed via the theoretical approach, to exalt the hydrogen evolution reaction in acidic medium. This topic is, certainly, of great interest and significance in the field of electrocatalysis. Therefore, its acceptance/publication by Nature Communications is strongly suggested after some minor concerns are addressed.

1. From an academic point of view, the use of non-sustainable materials (very expensive) is adequate. However, the authors should succinctly justify the use of rhodium. In other words, it would be possible to use the same procedure with abundant (non-PGM) elements?
2. What is the physical-chemical ingredient that makes P-Tri-RhO₂, after different treatments, present an excellent chemical stability?
3. The electrocatalytic data (e.g., Table S4) compares the activity of the materials by mass. This approach may be good, if all tested materials have similar particle size. The reviewer is concerned that this is not the case, as the data could be biased by particle size. Therefore, it is necessary to deliver comparative data using the intrinsic activity of the active data center based on its spin frequency (TOF).
5. Is there any reason to have used sulfuric acid instead of perchlorate acid?

6. Elementary question: Was a Pt electrode used to calibrate the SCE or Rh (cf. Fig S5)?

7. Regarding the origin of an exalted HER electrocatalyst, could the in situ generated hexagonal array of rhodium nanocrystals on the metastable two-dimensional substrate (Rh-NA/RhO₂) lead to a strong interaction with the substrate?

The reviewer suggests that the authors carefully review the XPS spectra of metallic Rh (Fig. S11) Rh-NA/RhO₂ versus Rh/C and the Hupd-deposited hydrogen oxidation peak (e.g., Fig S18).

Reviewer #3 (Remarks to the Author):

The authors claim that they have successfully fabricated Rh nanoparticle ordered arrays on a metastable RhO₂ surface. The Rh/RhO₂ composite shows superior catalytic performance in the hydrogen evolution reaction. Overall, the catalytic performance might be interesting. However, I have some serious concerns regarding their material characterizations, which may have misled the authors to a wrong conclusion.

Their major evidence of the Rh nanoparticle arrays is the presence of hexagonal superlattice under STEM observation. They ruled out the possibility of Moire pattern formation between RhO₂ and RhO₂ layers. It should be noted that the structure was partially reduced to form Rh, which exhibits an fcc structure according to the XRD pattern. The Rh (111) surface has close packed hexagonal Rh atom arrays with Rh-Rh distance between 0.27-0.28 nm. The superposition of Rh and RhO₂ layers will very likely generate a Moire pattern with periodicity of ~3 nm. Therefore, I suspect that the final Rh/RhO₂ composite might be actually Rh thin layer patches on top of RhO₂ surface, rather than ordered Rh nanoparticle arrays.

In general, it is dangerous to draw conclusions on 3D morphology based on simple 2D projections. The authors should consider STM, AFM, and other surface techniques to characterize their Rh/RhO₂ samples.

Some minor issues:

Standard XRD pattern of trigonal RhO₂ should be provided.

The average thickness of RhO₂ might be 5-10 nm based on 0001 XRD peak broadening. It is not rigorous to claim the thickness based on one thinnest area.

The language should be significantly polished before they submit the revised manuscript to another journal.

Response to Reviewers

Thank you for your precious time to constructive comments on our manuscript titled “**Unique Coupling of Nanocrystal Hexagonal Array and Two-Dimensional Metastable Substrate Boosts H₂-Production**” (Manuscript ID: NCOMMS-22-24345-T) for **Nature Communications**. We sincerely appreciate your comments and suggestions on our work, which are highly important for the further improvements to our manuscript. According to all the comments, we have made a detailed response and substantial revisions in our revised manuscript.

Reviewer #1 (Remarks to the Author):

This study describes the synthesis and fine characterization of a heterostructure comprising a metastable RhO₂ phase on which metallic rhodium particles have been formed. The proposed electrochemical method offers fine control of rhodium particle size and interparticle distance. The formed interface also enables H-spillover. All of these features combined result in excellent HER performance.

The proposed results are of interest to a broad community and may be published in this journal once the following changes have been made:

[Author’s Response]: We would like to thank you for the positive comments and recommendation of the publication in the Nature Communications. Your comments lead to further improve the quality of our work. We have modified our manuscript according to your valuable comments.

1) the introduction of the article must be completely rewritten. The ins and outs do not appear clearly and the scientific approach followed is very confused. The English in the introduction also needs to be reviewed.

[Author’s Response]: Following your suggestion, we have rewritten introduction and the English has been polished by Springer Nature Author Services.

[Added content]: [Page 2, Line 14] “The development of highly efficient electrochemical catalysts for the hydrogen evolution reaction (HER) through water splitting is a critical step in the advancement of hydrogen production for energy storage and conversion in modern industry¹⁻⁶. Furthermore, the simple HER is a less sophisticated process in terms of understanding the mechanism of the water catalytic reaction and the relationship between the electrocatalytic activity and crystal structure at the nanoscale than the four-step oxygen evolution reaction / oxygen reduction reaction processes. According to Trassati’s volcano plot, rhodium (Rh) or Rh-based materials are promising catalysts for the HER⁷⁻⁹. However, the adsorption energy of hydrogen (ΔG_{H}) on the Rh surface is still relatively high, which is unfavourable for the formation of H₂. In addition, the poor durability of these materials makes it necessary to design new structures to achieve enhanced HER performance⁹.”

[Page 3, Line 4] “It is well known that nanosized entities in periodic identical arrays strongly influence the electronic and transport properties of the material, providing collective characteristics different from those of the corresponding bulk structures^{10,11}. To date, many periodic nanostructures have been reported, showing promise in applications in energy conversion, catalysis, and photoelectronic devices¹²⁻¹⁷. Notably, a perfectly aligned nanocrystal array with an interparticle distance of a few nanometres may provide a new platform for pursuing unprecedented catalytic properties. However, the traditional nanolithography and template methods employed to fabricate the aligned array assembly always yield interparticle distances larger than 10 nm¹⁸. Thus, developing a new strategy to fabricate a perfectly aligned nanocrystal array with a short interparticle distance (less than 5 nm) is highly desirable.”

[Page 3, Line 13] “Two-dimensional (2D) metastable metal oxides may provide an ideal substrate

for overcoming the above challenges. 2D materials have attracted extensive attention due to their maximum atomic utilization, ideal activities and desirable durability¹⁹⁻³². In addition, metastable metal oxides provide extensive possibilities for synthesizing new interfacial structures due to their intrinsic metastable properties³³⁻³⁵. Furthermore, in the metal/oxide catalytic interfacial system, the hydrogen spillover effect can cause the activated hydrogen atoms to migrate from a hydrogen-rich area to a hydrogen-poor area, which may provide an effective way to further improve the HER activity^{36,37}.”

2) a surface characterization of the Rh-NA/RhO₂ must be given. In particular the preparation of this material involves the use of an alcohol and nafion. The presence of these organic species on the surface of the material must be probed (XPS C1s, O1s, F1s; FTIR). The possible presence of such species on the reaction should, if confirmed, be discussed.

[Author’s Response]: Following your valuable suggestion, we have added the XPS and FTIR spectra to probe Nafion and isopropanol on the catalyst. The XPS (C1s, O1s and F1s) and FTIR spectra of Rh-NA/RhO₂ were collected after long-term HER stability test and shown in **Supplementary Fig. 24**. The C-F, CH₃ and O-H bonds can be clearly observed from FTIR and XPS spectra, proving for the existence of Nafion and isopropanol in Rh-NA/RhO₂ electrocatalyst [Kosseoglou, D., Kokkinofa, R. & Sazou, D. FTIR spectroscopic characterization of NafionA (R)-polyaniline composite films employed for the corrosion control of stainless steel. *J. Solid. State. Electr.* **15**, 2619-2631 (2011); del Arco, M., Gutierrez, S., Martin, C. & Rives, V. FTIR study of isopropanol reactivity on calcined layered double hydroxides. *Phys. Chem. Chem. Phys.* **3**, 119-126 (2001)].

In this work, isopropanol is served as a dispersant agent. Nafion is served as dispersant and binding ones and affects the proton transfer. Both of them are important for the dispersity, stability and proton conduction of the electrocatalysts during the HER process [McGovern, M. S., Garnett, E. C., Rice, C., Masel, R. L. & Wieckowski, A. Effects of Nafion as a binding agent for unsupported nanoparticle catalysts. *J. Power Sources* **115**, 35-39 (2003); Lyckfeldt, O. Palmvist, L. & Carlström, E. Stabilization of alumina with polyelectrolyte and comb copolymer in solvent mixtures of water and alcohols. *J. Eur. Ceram. Soc.* **29**, 1069-1076 (2009), Ngo, T. T., Yu, T. L. & Lin, H. L. Influence of the composition of isopropyl alcohol/water mixture solvents in catalyst ink solutions on proton exchange membrane fuel cell performance. *J. Power Sources* **225**, 293-303 (2013)].

Supplementary Figure 24. XPS spectra of (a) C 1s, (b) F 1s and (c) O 1s for Rh-NA/RhO₂ after long-term HER stability test. (d) FTIR spectrum of Rh-NA/RhO₂ after long-term HER stability test.

[Added content]: [Page 13, Line 1] “XPS (C1s, O1s and F1s) and FTIR spectra of Rh-NA/RhO₂ were collected after the long-term HER stability test and are shown in **Supplementary Fig. 24**. The C-F, CH₃ and O-H bonds can be clearly observed from FTIR and XPS spectra, proving the existence of Nafion and isopropanol in the Rh-NA/RhO₂ electrocatalyst^{47,48}.”

[Page 18, Line 14] “In this work, isopropanol is served as a dispersant agent. Nafion is served as dispersant and binding ones and affects the proton transfer. Both of them are important for the dispersity, stability and proton conduction of the electrocatalysts during the HER process.”

47. Kosseoglou, D., Kokkinofa, R. & Sazou, D. FTIR spectroscopic characterization of NafionA (R)-polyaniline composite films employed for the corrosion control of stainless steel. *J. Solid. State. Electr.* **15**, 2619-2631 (2011).

48. del Arco, M., Gutierrez, S., Martin, C. & Rives, V. FTIR study of isopropanol reactivity on calcined layered double hydroxides. *Phys. Chem. Chem. Phys.* **3**, 119-126 (2001).

Supplementary Figure 24. XPS spectra of (a) C 1s, (b) F 1s and (c) O 1s for Rh-NA/RhO₂ after long-term HER stability test. (d) FTIR spectrum of Rh-NA/RhO₂ after long-term HER stability test.

Reviewer #2 (Remarks to the Author):

Processes at electrochemical interfaces are complex. It is thus imperative to stress, that the electrochemical interface is the place where the formation of surface chemical complexes formation can be induced by the processes of adsorption, and charge transfer (usually multi-electronic). In this context, the hydrogen evolution reaction (HER) is a process of paramount importance, via water splitting, to generate H₂, a clean fuel. The rate determining step (rds) of complex electrochemical processes are changeable with different experimental conditions. Even at different intensity of polarizations, the rds of HER are different as the observed Tafel slopes of ~30 mV/decade at weak polarization and of ~120 mV at strong polarization. In a whole, electrocatalysis is very challenging in mechanisms and kinetics.

The present manuscript by Fan et al, illustrates the important role that a metastable material can play for the design of advanced materials. This can be categorized as a noteworthy result. The strategy requires the fabrication of well-ordered nanocrystal matrix with sub-nanometer spacing. Therefore, it is evident that the synthesis of electrocatalytic materials with molecular-scale distance and well-aligned structure could be of great utility for the development of a catalytic system for hydrogen evolution. In this regard, the authors discuss several possibilities by showing results on an array of well-ordered nanocrystals in a pristine 2D metastable rhodium oxide (P-Tri-RhO₂). Apparently, the unique interaction between the nanocrystal array and the 2D metastable substrate allows the spillover of hydrogen atoms, as discussed via the theoretical approach, to exalt the hydrogen evolution reaction in acidic medium. This topic is, certainly, of great interest and significance in the field of electrocatalysis. Therefore, its acceptance/publication by Nature Communications is strongly suggested after some minor concerns are addressed.

[Author's Response]: We would like to thank you very much for the positive comments and strong recommendation of the publication in Nature Communications. We clarify all comments one-by-one below. We have modified our manuscript based on your comments accordingly and all modifications have been highlighted in the revised manuscript. We have realized that your comments and suggestions lead to improve the quality of our work.

1. From an academic point of view, the use of non-sustainable materials (very expensive) is adequate. However, the authors should succinctly justify the use of rhodium. In other words, it would be possible to use the same procedure with abundant (non-PGM) elements?

[Author's Response]: This is a relevant question. First, we justify the use of rhodium in this work. Hydrogen adsorption free energy (ΔG_H) on catalyst is considered as an indicator of HER activity. Making the ΔG_H closer to be the "volcano" top is the main target for designing high performance HER catalyst. As shown in **Fig. R1**, the Trassati's volcano plot shows that Rhodium (Rh) or Rh-based materials are the promising candidates for HER [Quaino, P., Juarez, F., Santos, E. & Schmickler, W. Volcano plots in hydrogen electrocatalysis-uses and abuses. *Beilstein J. Nanotechnol.* **5**, 846-854 (2014)]. However, the adsorption energy of hydrogen on the Rh surface is still relatively high, making it difficult for the desorption of H₂. Therefore, it is necessary to design new structure in order to achieve the enhanced HER performance. Moreover, one important aspect that should be considered is that the poor chemical stability of non-PGM-based catalysts in acid media, which makes it important to develop PGM catalysts in acidic media.

In addition, according to our work, the metastable trigonal phase RhO₂ is important for the formation of Rh nanocrystal array. First of all, the mismatch between face-centered cubic Rh and P-Tri-RhO₂

was determined to be only 0.45% $((0.1345-0.1339) \times 100\% / 0.1339 = 0.45\%)$ (**Supplementary Fig. 15**), indicating it feasible for the *in-situ* epitaxial growth of face-centered cubic Rh on the P-Tri-RhO₂ substrate. For a sharp comparison, the mismatch between face-centered cubic Rh and Rutile-RhO₂ can reach 15.3% $((0.44862-0.38004) \times 100\% / 0.44862 = 15.3\%)$, indicating it is difficult to obtain Rh nanocrystal array by using Rutile-RhO₂ as substrate. We also conducted the same *in-situ* electrochemical reduction step for Rutile-RhO₂ and the corresponding results are shown in **Supplementary Fig. 14**. No Rh nanocrystal arrays are observed, suggesting the key role of metastable two-dimensional P-Tri-RhO₂ as a substrate on forming this special array structure. Therefore developing metastable non-PGM oxide may provide promising candidates for constructing the nanocrystal array structures. We have put a lot of efforts to use the same procedure with non-PGM precursors, but it is impossible to obtain the metastable non-PGM oxides.

[redacted]

[Added content]: [Page 2, Line 19] “According to Trassati’s volcano plot, rhodium (Rh) or Rh-based materials are promising catalysts for the HER⁷⁻⁹. However, the adsorption energy of hydrogen (ΔG_H) on the Rh surface is still relatively high, which is unfavourable for the formation of H₂. In addition, the poor durability of these materials makes it necessary to design new structures to achieve enhanced HER performance⁹.”

7. Quaino, P., Juarez, F., Santos, E. & Schmickler, W. Volcano plots in hydrogen electrocatalysis—uses and abuses. *Beilstein J. Nanotechnol.* **5**, 846-854 (2014).
8. Costentin, C. & Saveant, J. M. Towards an intelligent design of molecular electrocatalysts. *Nat. Rev. Chem.* **1**, 0087 (2017).
9. Zhu, L. L. et al. A rhodium/silicon co-electrocatalyst design concept to surpass platinum hydrogen evolution activity at high overpotentials. *Nat. Commun.* **7**, 12272 (2016).

2. What is the physical-chemical ingredient that makes P-Tri-RhO₂, after different treatments, present an excellent chemical stability?

[Author’s Response]: Thank you for your comment. The edge-sharing structure of P-Tri-RhO₂ is important for the excellent stability. Compared to corner-sharing 2D materials, P-Tri-RhO₂ with edge-sharing structures possess one more couple of connected bonds, which lead to edge-sharing structures may be more stable than corner-sharing ones in theory [Guan, D. Q. et al. Exceptionally robust face-sharing motifs enable efficient and durable water oxidation. *Adv. Mater.* **33**, 2103392 (2021); Lin, X. et al. 5f covalency synergistically boosting oxygen evolution of UCoO₄ catalyst. *J.*

Am. Chem. Soc. **144**, 416-423 (2022); Yang, L. et al. Efficient oxygen evolution electrocatalysis in acid by a perovskite with face-sharing IrO₆ octahedral dimers. *Nat. Commun.* **9**, 5236 (2018)].

3. The electrocatalytic data (e.g., Table S4) compares the activity of the materials by mass. This approach may be good, if all tested materials have similar particle size. The reviewer is concerned that this is not the case, as the data could be biased by particle size. Therefore, it is necessary to deliver comparative data using the intrinsic activity of the active data center based on its spin frequency (TOF).

[Author's Response]: Thank you for your valuable suggestion. Following your suggestion, we calculated the TOF to compare the intrinsic activities of Rh-NA/RhO₂, Rutile-RhO₂, C-Rh/C and Pt/C. As shown in **Supplementary Fig. 22a**, Rh-NA/RhO₂ has a higher TOF than those of Rutile-RhO₂, C-Rh/C and Pt/C at different potentials (vs. RHE). In details, a TOF of 4.68 s⁻¹ was achieved by Rh-NA/RhO₂ at the overpotential of 20 mV, 17.3, 18 and 6.5 times higher than those of Rutile-RhO₂ (0.27 s⁻¹), C-Rh/C (0.26 s⁻¹) and Pt/C (0.72 s⁻¹) (**Supplementary Fig. 22a**), indicating the excellent HER intrinsic activity of Rh-NA/RhO₂.

Turnover frequencies (TOFs) of electrocatalysts were calculated as follows:

$$TOF = \frac{3.12 \times 10^{15} \frac{H_2/s}{cm^2} per \frac{mA}{cm^2} \times |j|}{active\ sites} \quad (\text{Equation 1})$$

The active sites of electrocatalysts were calculated as follows:

$$Active\ sites = \frac{Q}{1.602 \times 10^{-19}} \quad (\text{Equation 2})$$

$$Q = \frac{S_{peak}}{\nu} \quad (\text{Equation 3})$$

Where S_{peak} is the integral area of adsorbed hydrogen desorption peak in the CV curve (**Supplementary Fig. 21**), ν is the scan rate of 50 mV s⁻¹ and Q is quantity of electric charge.

Supplementary Figure 22. (a) The comparison of TOFs for Rh-NA/RhO₂, Rutile-RhO₂, C-Rh/C and Pt/C at different potentials (vs. RHE). (b) The TOFs of Rh-NA/RhO₂, Rutile-RhO₂, C-Rh/C and Pt/C at the overpotential of 20 mV.

[Added content]: [Page 12, Line 5] “To further eliminate the effect of particle size and surface area, the turnover frequencies (TOFs) were calculated to compare the catalytic activities of different catalysts. As shown in **Supplementary Fig. 22a**, Rh-NA/RhO₂ always has a higher TOF than Rutile-RhO₂, C-Rh/C and Pt/C at different potentials (vs. RHE). In detail, a TOF of 4.68 s⁻¹ is

achieved with Rh-NA/RhO₂ at overpotentials of 20 mV, 17.3, 18 and 6.5 times higher than those of Rutile-RhO₂ (0.27 s⁻¹), C-Rh/C (0.26 s⁻¹) and Pt/C (0.72 s⁻¹) (**Supplementary Fig. 22b**), indicating the excellent HER intrinsic activity of Rh-NA/RhO₂.”

[Page 19, Line 2] “Turnover frequencies (TOFs) of electrocatalysts were calculated as follows:

$$TOF = \frac{3.12 \times 10^{15} \frac{H_2/s}{cm^2} per \frac{mA}{cm^2} \times |j|}{\text{active sites}} \quad (\text{Equation 1})$$

The active sites of electrocatalysts were calculated as follows:

$$\text{Active sites} = \frac{Q}{1.602 \times 10^{-19}} \quad (\text{Equation 2})$$

$$Q = \frac{S_{\text{peak}}}{\nu} \quad (\text{Equation 3})$$

Where S_{peak} is the integral area of adsorbed hydrogen desorption peak in the CV curve (**Supplementary Fig. 21**), ν is the scan rate of 50 mV s⁻¹ and Q is quantity of electric charge.”

Supplementary Figure 22. (a) The comparison of TOFs for Rh-NA/RhO₂, Rutile-RhO₂, C-Rh/C and Pt/C at different potentials (vs. RHE). (b) The TOFs of Rh-NA/RhO₂, Rutile-RhO₂, C-Rh/C and Pt/C at overpotential of 0.1 V.

4. Is there any reason to have used sulfuric acid instead of perchlorate acid?

[Author’s Response]: Thank you for your question. In industrial production, sulfuric acid solution is usually used as electrolytes rather than perchloric acid solutions. In addition, compared with sulfuric acid solution, perchloric acid solution is more volatile, which may lead to the obvious performance degradation during electrocatalytic stability test. Therefore, we selected sulfuric acid solution as electrolyte rather than perchlorate acid solution for hydrogen evolution reaction in this work.

5. Elementary question: Was a Pt electrode used to calibrate the SCE or Rh (cf. Fig S15)?

[Author’s Response]: Thank you for your question. In this work, two Pt electrodes were used as the working and counter electrodes to calibrate the SCE rather than Rh.

[Added content]: [Page 11, Line 2] “Before electrochemical tests, two Pt electrodes were used as the working and counter electrodes to calibrate the SCE.”

6. Regarding the origin of an exalted HER electrocatalyst, could the in situ generated hexagonal array of rhodium nanocrystals on the metastable two-dimensional substrate (Rh-NA/RhO₂) lead to a

strong interaction with the substrate?

The reviewer suggests that the authors carefully review the XPS spectra of metallic Rh (Fig. S11) Rh-NA/RhO₂ versus Rh/C and the H_{upd}-deposited hydrogen oxidation peak (e.g., Fig S18).

[Author's Response]: Thank you for your valuable suggestion. The XPS spectra of Rh 3d for Rh-NA/RhO₂ and Rh/C are shown in **Supplementary Fig. 12**. Compared with Rh/C, the binding energy of Rh⁰ in Rh-NA/RhO₂ shifted to higher binding energy by about 0.15 eV, indicating the existence of strong electronic interaction between Rh and RhO₂ [Li, Z. et al. Stable rhodium (IV) oxide for alkaline hydrogen evolution reaction. *Adv. Mater.* **32**, 1908521 (2020), Bai, S. X. et al. Surface engineering of RhOOH nanosheets promotes hydrogen evolution in alkaline. *Nano energy* **78**, 105224 (2020)].

As suggested, we also compared the CVs of Rh-NA/RhO₂ and C-Rh/C in H₂-saturated 0.5 M H₂SO₄. As shown in **Fig. R2**, one can see that an obvious left shift of hydrogen desorption peak was observed at CVs, indicating Rh-NA/RhO₂ is easier to desorb hydrogen than this of C-Rh/C.

Supplementary Figure 12. The XPS spectra of Rh 3d peaks for Rh-NA/RhO₂ and Rh/C.

Figure R2. The CVs of Rh-NA/RhO₂ and C-Rh/C in H₂-saturated 0.5 M H₂SO₄.

[Added content]: [Page 10, Line 9] “Compared with C-Rh/C, the binding energy of Rh in Rh-NA/RhO₂ shifts to a higher binding energy by approximately 0.15 eV, indicating the existence of a strong electronic interaction between the Rh nanocrystal array and the P-Tri-RhO₂ substrate⁴³.”

Supplementary Figure 12. The XPS spectra of Rh 3d peaks for Rh-NA/RhO₂ and Rh/C.

Reviewer #3 (Remarks to the Author):

The authors claim that they have successfully fabricated Rh nanoparticle ordered arrays on a metastable RhO₂ surface. The Rh/RhO₂ composite shows superior catalytic performance in the hydrogen evolution reaction. Overall, the catalytic performance might be interesting. However, I have some serious concerns regarding their material characterizations, which may have misled the authors to a wrong conclusion.

Their major evidence of the Rh nanoparticle arrays is the presence of hexagonal superlattice under STEM observation. They ruled out the possibility of Moire pattern formation between RhO₂ and RhO₂ layers. It should be noted that the structure was partially reduced to form Rh, which exhibits an fcc structure according to the XRD pattern. The Rh (111) surface has close packed hexagonal Rh atom arrays with Rh-Rh distance between 0.27-0.28 nm. The superposition of Rh and RhO₂ layers will very likely generate a Moire pattern with periodicity of ~3 nm. Therefore, I suspect that the final Rh/RhO₂ composite might be actually Rh thin layer patches on top of RhO₂ surface, rather than ordered Rh nanoparticle arrays.

In general, it is dangerous to draw conclusions on 3D morphology based on simple 2D projections. The authors should consider STM, AFM, and other surface techniques to characterize their Rh/RhO₂ samples.

[Author's Response]: Thank you for your valuable comments. We fully agree with reviewer#3 that one cannot simply draw conclusions from 3D morphology based on simple 2D projections and we also agree with reviewer#3 that the patterns of nanocrystal arrays are similar with Rh-RhO₂ Moire patterns at the first glance. However, our conclusion was obtained after the carefully theoretical simulations of all possible 3D morphologies, then make 2D projections to compare with our experimental results. In this way, we have safely excluded Moire patterns.

1. First of all, we calculated the surface density (SD, the number of Rh atoms per nm²) of Rh atoms for Moire pattern obtained from one layer P-Tri-RhO₂ and one layer metallic Rh and the nanocrystal arrays in our work (Rh-NA/RhO₂). As shown in **Supplementary Note 1**, it shows the maximum SD of Rh atoms is 28.055 nm⁻² for the Moire pattern consisting of one single-layer P-Tri-RhO₂ and one single-layer metallic Rh, much lower than this of Rh-NA/RhO₂ (47.3±1.2 nm⁻²), indicating this unique structure is Rh nanocrystal arrays instead of Moire pattern (**Fig. R3**).
2. We also simulated the Moire pattern by rotating one single-layer P-Tri-RhO₂ and one single-layer metallic Rh. As shown in **Supplementary Fig. 16**, a typical Moire pattern was obtained by twisting one single-layer P-Tri-RhO₂ and one single-layer metallic Rh with the rotation angle of 3°. The atomic enlarged areas of this Moire pattern are completely different from Rh nanocrystal array (**Supplementary Fig. 16b,c**), indicating Rh nanocrystal array is real rather than Moire pattern (**Fig. 2f, 2g**).
3. It is very difficult to grow thin metallic Rh layer on Tri-RhO₂. The growth of surface layer structure depends on the differences of surface energies between cover material and substrate material. When the surface energy of the cover material is less than that of the substrate, the continuous surface may grow on the substrate. On the contrary, the growth of materials cannot become continuous [Xue, M. S., Guo, J. D. & Guo, Q. L. Effect of polar surface on the growth of Au. *RSC Adv.* **5**, 11109 (2015)]. P-Tri-RhO₂ is a typical layered material, where the interaction between layers is the weak Van der Waal's force. Therefore, the surface energy of P-Tri-RhO₂ is much lower than that of metallic Rh. Therefore, it is hard to grow continuous thin metallic Rh layer on the surface of P-Tri-RhO₂.

4. Furthermore, we also simulate the XRD pattern of different atomic layers Rh. As shown in **Supplementary Fig. 17**, the simulated XRD peaks of different atomic layers Rh cannot be detected in Rh-NA/RhO₂, excluding the thin Rh atomic layer on P-Tri-RhO₂.

Figure R3. The theoretical maximum SD for Moire pattern obtained from one layer P-Tri-RhO₂ and one layer metallic Rh. In addition, the actual SD of Rh-NA/RhO₂ was determined to be $47.3 \pm 1.2 \text{ nm}^{-2}$.

Supplementary Figure 16. Simulated Moire pattern was obtained by twisting one single-layer P-Tri-RhO₂ and one single-layer metallic Rh with the rotation angle of 3°. (b, c) The enlarged areas from (a).

Supplementary Figure 17. The XRD pattern of Rh-NA/RhO₂ and the simulated XRD patterns of different layers Rh.

[Added content]: [Page 9, Line 19] “We also simulated a Moire pattern by rotating a single layer of P-Tri-RhO₂ and a single layer of metallic Rh. As shown in **Supplementary Fig. 16**, a typical Moire pattern is obtained by twisting a single layer of P-Tri-RhO₂ and a single layer of metallic Rh with a rotation angle of 3°. The atomic enlarged areas of this Moire pattern are completely different from those of the Rh nanocrystal array (**Fig. 2f, 2g**), indicating that the Rh nanocrystal array is real rather than a Moire pattern. Furthermore, we also simulated the XRD patterns of different atomic layers of Rh. As shown in **Supplementary Fig. 17**, the simulated XRD peaks of different atomic layers of Rh cannot be detected in Rh-NA/RhO₂, excluding a thin Rh atomic layer on P-Tri-RhO₂.”

Supplementary Figure 16. Simulated Moire pattern was obtained by twisting one single-layer P-Tri-RhO₂ and one single-layer metallic Rh with the rotation angle of 3°. (b, c) The enlarged areas from (a).

Supplementary Figure 17. The XRD pattern of Rh-NA/RhO₂ and the simulated XRD patterns of different layers Rh.

Some minor issues:

Standard XRD pattern of trigonal RhO₂ should be provided.

[Author's Response]: Thank you for your valuable suggestion. As suggested, we provide the standard XRD pattern of P-Tri-RhO₂. As shown in **Supplementary Fig. 3**, the simulated XRD pattern of P-Tri-RhO₂ is almost same with XRD pattern of P-Tri-RhO₂, further confirming its trigonal phase.

Supplementary Figure 3. Comparison of XRD patterns of P-Tri-RhO₂ (black curve) and simulated X-ray diffraction peak of P-Tri-RhO₂ (red curve).

[Added content]: [Page 5, Line 2] “In addition, the simulated XRD pattern of P-Tri-RhO₂ is almost the same as the XRD pattern of P-Tri-RhO₂, further confirming its trigonal phase (**Supplementary Fig. 3**).”

Supplementary Figure 3. Comparison of XRD patterns of P-Tri-RhO₂ (black curve) and simulated X-ray diffraction peak of P-Tri-RhO₂ (red curve).

The average thickness of RhO₂ might be 5-10 nm based on 0001 XRD peak broadening. It is not rigorous to claim the thickness based on one thinnest area.

[Author’s Response]: Thank you for your comment. P-Tri-RhO₂ is a typical layered material. The interaction between the layers is the weak Van der Waal’s force, which makes it easy to be exfoliated during ultrasonic treatment in the AFM sample preparation. As shown in **Fig. R4a**, the thickness of the P-Tri-RhO₂ before ultrasonic treatment was determined to be 7.0 nm, which is almost consistent with XRD result. However, the thickness of P-Tri-RhO₂ after ultrasonic treatment was determined to be only 1.1 nm (**Fig. R4b**).

Figure R4. AFM images of P-Tri-RhO₂. (a) AFM image of P-Tri-RhO₂ before exfoliation, inset is the corresponding height profile, (b) AFM image of P-Tri-RhO₂ after exfoliation, inset is the corresponding height profile.

The language should be significantly polished before they submit the revised manuscript to another journal.

[Author's Response]: Thank you for your comment. Following your suggestion, the new version of manuscript now is polished by Springer Nature Author Services.

REVIEWER COMMENTS

Reviewer #2 (Remarks to the Author):

The authors have adequately revised their manuscript taking into account my questions (as reviewer #2) as well as those raised by reviewers #1 and #3. And because of these improvements, I am pleased to recommend publication in Nature Communications.

Reviewer #3 (Remarks to the Author):

It seems that the authors made the main conclusion on the formation of ordered array of Rh nanoparticles mostly by simulating Moire patterns. In my previous comments, I made one suggestion on the Moire patterns. There could be many possibilities for Moire pattern formation by varying the orientation of Rh and RhO₂ structures (not just tilting along one axis).

Growing Rh thin layer on RhO₂ may be difficult. But it does not mean impossible.

On the other hand, there is no driving force for the formation of ORDERED array instead of random pattern of Rh nanoparticles.

Again, it is very risky to claim such a unique structure (ordered array) without showing any 3D evidence. The authors should attempt to conduct AFM, STM, or high-resolution SEM or STEM to confirm the ordered arrays. It will certainly attract a lot of interest if the claimed structure is true.

Response to Reviewers

The reviewer' comments are shown in black and our responses are shown in blue. The changes in the manuscript are indicated in red.

Reviewer #2 (Remarks to the Author):

The authors have adequately revised their manuscript taking into account my questions (as reviewer #2) as well as those raised by reviewers #1 and #3. And because of these improvements, I am pleased to recommend publication in Nature Communications.

[Author's Response]: We would like to thank you very much for the recommendation of the publication in Nature Communications.

Reviewer #3 (Remarks to the Author):

It seems that the authors made the main conclusion on the formation of ordered array of Rh nanoparticles mostly by simulating Moire patterns. In my previous comments, I made one suggestion on the Moire patterns. There could be many possibilities for Moire pattern formation by varying the orientation of Rh and RhO₂ structures (not just tilting along one axis). Growing Rh thin layer on RhO₂ may be difficult. But it does not mean impossible.

On the other hand, there is no driving force for the formation of ORDERED array instead of random pattern of Rh nanoparticles.

Again, it is very risky to claim such a unique structure (ordered array) without showing any 3D evidence. The authors should attempt to conduct AFM, STM, or high-resolution SEM or STEM to confirm the ordered arrays. It will certainly attract a lot of interest if the claimed structure is true.

[Author's Response]: We would like to thank the reviewer#3 for the precious time to constructive suggestions on our manuscript, which are important for further improvements of our manuscript.

Following the suggestion of reviewer#3, we have now included the high-angle annular dark-field imaging-scanning transmission electron microscopy (HAADF-STEM) and the energy dispersive X-ray spectroscopy (EDX) line scanning of Rh-NA/RhO₂, which we have obtained in the last months. The corresponding results are shown in **Supplementary Fig. 17**. The Rh nanocrystal arrays are clearly observed in the HAADF-STEM image (**Supplementary Fig. 17a**). The EDX line scanning profile in **Supplementary Fig. 17b** shows that the contrasts on the Rh and O elements demonstrate nearly equal spaced Rh nanocrystals in a consistent way. Although the long-term irradiation of the high energy electron beam during the STEM characterization causes the slight distortion of nanocrystal array, the results can confirm the formation of ordered nanocrystal array instead of the Morie pattern. We would like to thank the reviewer#3 again for the suggestions to make our manuscript more interesting.

Supplementary Figure 17. (a) The HAADF-STEM image and (b) the corresponding EDX line

scanning profile of Rh-NA/RhO₂.

[Added content]: [Page 10, Line 4] “The Rh nanocrystal arrays are clearly observed in the HAADF-TEM image (Supplementary Fig. 17a). The EDX line scanning profile in Supplementary Fig. 17b shows that the contrasts on the Rh and O elements demonstrate nearly equal spaced Rh nanocrystals in a consistent way, which also excludes the formation of Moire pattern.”

Supplementary Figure 17. (a) The HAADF-STEM image and (b) the corresponding EDX line scanning profile of Rh-NA/RhO₂.

Concerning the point: “There could be many possibilities for Moire pattern formation by varying the orientation of Rh and RhO₂ structures (not just tilting along one axis).”

This statement is only correct without considering detailed symmetry of our Rh nanoarray and RhO₂ substrate, which contains a 3-fold symmetry. In this case, one needs a 3-fold symmetry for both Rh and RhO₂ lattice limiting the possibility of orientation, therefore all possible Moire patterns can be safely excluded by our simulations.

Concerning another point: “there is no driving force for the formation of ORDERED array instead of random pattern of Rh nanoparticles.”

First, growing ordered metal nanoarrays on an oxide substrate was reported by Hamm, G. *et al.*, where they deposited regular two-dimensional array of bimetallic Au-Pd nanoparticles on alumina template [Hamm, G. *et al. Nanotechnology* **17**, 1943 (2006)].

Another example is the fabrication of anodic aluminum oxide (AAO), as a typical self-ordered hexagonal nanoarrays formed by anodization of aluminum plates [Masuda, H. *et al. Science* **268**, 1466 (1995); Su, Z. *et al. Adv. Mater.* **20**, 3663 (2008); Runge, J. M. Anodic Aluminum Oxide Growth and Structure, *The Metallurgy of Anodizing Aluminum*, 281 Springer (2018)], although the real driving force for the AAO still remains unclear as pointed by Su, Z. *et al. Adv. Mater.* **20**, 3663 (2008) yet.

In Masuda’s work, a long period anodization at 40 V is applied for preparing AAO, while a constant potential of -0.4 V vs. RHE is applied for preparing Rh-NA/RhO₂ in our work.

REVIEWERS' COMMENTS

Reviewer #3 (Remarks to the Author):

With the new HAADF and EDX results, the conclusion is now more convincing. I would suggest the authors to include these figures in the main text rather than supporting information. Although the mechanism for the formation of ordered nanoparticle array is still unclear, at least they have confirmed the structure. Based on this, I would recommend acceptance of this manuscript. Good work.

Response to the Reviewer

The reviewer's comments are shown in black and our responses are shown in blue. The changes in the manuscript are indicated in red.

Reviewer #3 (Remarks to the Author):

With the new HAADF and EDX results, the conclusion is now more convincing. I would suggest the authors to include these figures in the main text rather than supporting information. Although the mechanism for the formation of ordered nanoparticle array is still unclear, at least they have confirmed the structure. Based on this, I would recommend acceptance of this manuscript. Good work.

[Author's Response]: We would like to thank you for the positive comments and recommendation of the publication in the Nature Communications. Your comments lead to further improve the quality of our work. According to your comments, we have added HAADF and EDX results into the main text and corresponding results are shown in **Fig. 2h, 2i**.

Figure 2. (h) The HAADF-STEM image and **(i)** the corresponding EDX line scanning profile of Rh-NA/RhO₂.

[Added content]:

Figure 2. (h) The HAADF-STEM image and **(i)** the corresponding EDX line scanning profile of Rh-NA/RhO₂.